# EVERY LANGUAGE MODEL HAS A FORGERY-RESISTANT SIGNATURE

**Matthew Finlayson, Xiang Ren, & Swabha Swayamdipta**
`{mfinlays,xiangren,swabhas}@usc.edu`
University of Southern California

## ABSTRACT

The ubiquity of closed-weight language models with public-facing APIs has generated interest in forensic methods, both for extracting hidden model details (e.g., parameters) and for identifying models by their outputs. One successful approach to these goals has been to exploit the geometric constraints imposed by the language model architecture and parameters. In this work, we show that a lesser-known geometric constraint—namely, that language model outputs lie on the surface of a high-dimensional ellipse—functions as a signature for the model and can be used to identify the source model of a given output. This *ellipse signature* has unique properties that distinguish it from existing model-output association methods like language model fingerprints. In particular, the signature is *hard to forge*: without direct access to model parameters, it is practically infeasible to produce log-probabilities (logprobs) on the ellipse using currently known methods. Secondly, the signature is *naturally occurring*, since all[1] language models have these elliptical constraints. Thirdly, the signature is *self-contained*, in that it is detectable without access to the model inputs or the full weights. Finally, the signature is *compact and redundant*, as it is independently detectable in each logprob output from the model. We evaluate a novel technique for extracting the ellipse from small models and discuss the practical hurdles that make it infeasible for production-scale models. Finally, we use ellipse signatures to propose a protocol for language model output verification, analogous to cryptographic symmetric-key message authentication systems.

## 1 INTRODUCTION

The proliferation of closed-weight language models has incentivized the development of methods for language model forensics, i.e., *post hoc* methods to learn about language models and their outputs with limited access. Recent work in this area has introduced methods to exploit linear constraints imposed by the model architecture to identify the source of generations through a limited API (Finlayson et al., 2024; Yang & Wu, 2024). These methods use model constraints as a type of *signature*, where one can verify that an output came from a specific language model by simply checking that the output (typically, the model's output log probability vector) satisfies the model's constraints. Thus the signature functions as a type of watermark or fingerprint (Xu et al., 2025b; Liu et al., 2024).

Another, lesser-known constraint can also function as a model signature: the constraint that model outputs lie on a high-dimensional ellipse (a hyperellipsoid) (Carlini et al., 2024, Appendix G). Model outputs can be verified by checking that they lie on the model ellipse. In this work, we explain the ellipse constraints and show how they can be used to identify the model that generated an output.

We find that ellipse signatures have a set of four unique properties, which differentiate them from previous signatures and other model identification methods. In combination, these properties fill a new niche in the landscape of output verification systems. We emphasize that signatures are not necessarily *better* than other methods, rather their properties make them more suitable in specific situations. First, ellipse signatures are *forgery-resistant*, i.e., the signature is hard to fake for closed-weight[2] models.

---

[1]More precisely, this applies to models with a final normalization layer, which encompass virtually all widely used language models today.

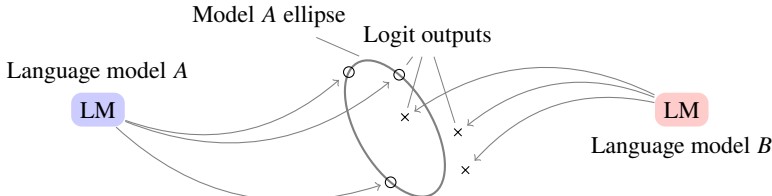

Figure 1: Language model logits are subject to constraints that force them to lie on a high-dimensional ellipse. This constitutes a *signature* because we can identify which model generated an output by checking which ellipse it lies on. Among other unique properties, we show that ellipse signatures are forgery-resistant because it is computationally hard in practice to generate signed logits without access to the model parameters.

This differentiates the ellipse signature from previously known linear signatures (Finlayson et al., 2024; Yang & Wu, 2024). Second, ellipse signatures are *naturally occurring*, because virtually all modern language models have a unique ellipse constraint and therefore sign their outputs. In contrast, many watermark and fingerprinting methods require the model or inference provider to proactively implement the system (e.g., Zeng et al., 2025; Cui et al., 2025). Third, ellipse signatures are *self-contained*: output detection does not require access to the model parameters or input. Self-containment is useful for situations where a provider wants a trusted third party to be able to verify outputs from their private language model without giving away the model parameters or prompt. Finally, the ellipse signature is *compact and redundant* because the signature is present and detectable in any single generation step.[3] Consequently, a single generation step is sufficient to identify the generating model. This property is unique because many existing output identification methods require outputs from multiple generation steps to gather evidence of the generator's identity (e.g., Kirchenbauer et al., 2023).

The most interesting and least obvious property of ellipse signatures is their forgery resistance (Section 3). We define forgery as generating logprobs that conform to the model constraints without direct access to the model parameters. Previously studied linear signatures (Finlayson et al., 2024; Ying et al., 2012) can be forged by extracting the linear constraints from the model API and generating logprobs to satisfy them. In comparison, ellipse extraction from API-protected language models is extremely difficult, and thus forgery is much harder. Ellipse extraction is difficult for at least two reasons: it is expensive, with $O(d^3 \log d)$ query complexity in an OpenAI-like API, and actually fitting the ellipse has $O(d^6)$ time complexity. We demonstrate these challenges by implementing a new ellipse-specific extraction method and testing it on small models. We are not aware of any ellipse forgery method that avoids having to fit the ellipse, though we cannot at this time mathematically rule out the possibility. For this reason, we adopt the term "forgery resistance" rather than "unforgeability".

The forgery resistance of ellipse signatures, coupled with the relative ease of verifying outputs on the ellipse, presents an intriguing opportunity for an output verification system. We propose a system analogous to cryptographic message authentication (Pass & Shelat, 2010), where the model ellipse functions as the secret key. Parties with access to secret language model parameters (including the ellipse parameters) can generate logprobs that can in turn be verified only by those who also have access to the secret ellipse (Section 4). We argue that such a system has implications for model forensics, regulation, and accountability for opaque choices made by language model providers.

## 2 LANGUAGE MODEL ELLIPSES ARE SIGNATURES

We set up the mathematical formulation of the language model ellipse and show how it functions as a signature. To begin, let us assume that our language model has a typical architecture, like the one shown in Figure 2. In particular, we are interested in models with a vocabulary size $v$ much larger than their hidden size $d$, that have a final sequence of layers consisting of a normalization, followed by a linear layer $\mathbb{R}^d \rightarrow \mathbb{R}^v$.

---

[2]As opposed to open-weight models with publicly-released parameters.

[3]By "single generation step", we mean the probabilities of every next-token candidate in the vocabulary.

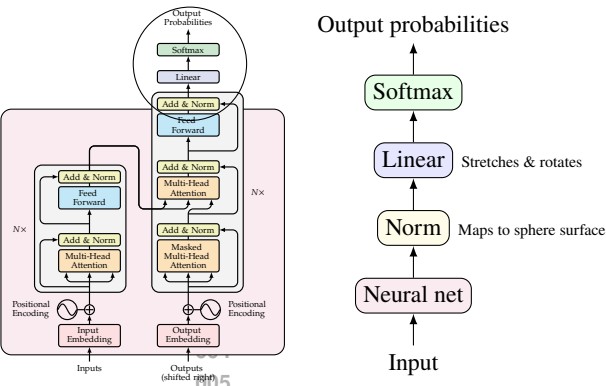

Figure 2: A typical language model's final layers consist of normalization followed by a linear (or affine) transformation. Normalization has the effect of mapping the representations onto the surface of a sphere, while the linear layer stretches and rotates this sphere, resulting in an ellipse. Transformer diagram credit Vaswani et al. (2017); Negrinho (2020).

## 2.1 LANGUAGE MODEL OUTPUTS LIE ON AN ELLIPSE

Language model outputs lie on an ellipse because the penultimate model layer normalizes the activations before projecting them linearly into $\mathbb{R}^v$. Model architects tend to choose one of two normalization schemes: the *root-mean-square* (RMS) norm (Zhang & Sennrich, 2019) or the *layer* norm (Ba et al., 2016). We will focus on the RMS norm because it is simpler, though we revisit the layer norm in Section E for completeness. The RMS norm is formally defined as

$$\text{RMSNorm}(\boldsymbol{x}) = \frac{\boldsymbol{x}}{\sqrt{\varepsilon + \mathbb{E}[\boldsymbol{x}^2]}}, \tag{1}$$

where $\varepsilon$ is a small, positive term added to prevent division by zero. We note that the magnitude of the normalized output is $\sqrt{d}$. For pedagogical reasons, we will proceed with a simplified, scaled RMS norm without an $\varepsilon$ term $\text{norm}(\boldsymbol{x}) = \boldsymbol{x}/\sqrt{d \cdot \mathbb{E}[\boldsymbol{x}^2]}$, so that the normalized outputs will have magnitude 1. Normalization has the property of mapping inputs onto the surface of a $d$-dimensional sphere because it sets the magnitude to 1 while preserving the direction. To simplify notation, we will denote $\text{norm}(\boldsymbol{x})$ as $\hat{\boldsymbol{x}}$. It is also common to apply a learned *element-wise affine* transformation after normalizing, by multiplying by $\boldsymbol{\gamma} \in \mathbb{R}^d$ then adding a bias term $\boldsymbol{\beta} \in \mathbb{R}^d$, so that the output is $\boldsymbol{\gamma} \odot \hat{\boldsymbol{x}} + \boldsymbol{\beta}$. We call the output of the normalization layer the *(final) hidden state* of the model.

To obtain a logit vector from the hidden state, the model multiplies the hidden state by the unembedding matrix $\boldsymbol{W}$. Because the normalized representations $\hat{\boldsymbol{x}}$ lie on a sphere, and $\boldsymbol{W}(\boldsymbol{\gamma} \odot \hat{\boldsymbol{x}} + \boldsymbol{\beta})$ is an affine transformation, the logits lie on the surface of a $d$-dimensional ellipse. It is important to note that the ellipse, like the sphere, is $d$-dimensional, even if it is projected into a $v$-dimensional space. The $d$-dimensional ellipse inhabits $\mathbb{R}^v$ in the same way that comet's 2D elliptical orbit inhabits 3D space.

Finally, language model APIs usually return log-probabilities (logprobs), i.e., $\log \text{softmax}(\boldsymbol{W}(\boldsymbol{\gamma} \odot \hat{\boldsymbol{x}} + \boldsymbol{\beta}))$, rather than logits. Since the softmax function is invariant to scalar addition, the logprobs remain unchanged if we assume that the logits are centered, i.e., $\boldsymbol{C}\boldsymbol{W}(\boldsymbol{\gamma} \odot \hat{\boldsymbol{x}} + \boldsymbol{\beta})$ where $\boldsymbol{C} = \boldsymbol{I} - \frac{1}{v}$. This simplifying assumption makes recovering logits from logprobs possible, since $\boldsymbol{C} \log \text{softmax}(\boldsymbol{W}(\boldsymbol{\gamma} \odot \hat{\boldsymbol{x}} + \boldsymbol{\beta})) = \boldsymbol{C}\boldsymbol{W}(\boldsymbol{\gamma} \odot \hat{\boldsymbol{x}} + \boldsymbol{\beta})$. Section A gives a more in-depth explanation of this assumption.

## 2.2 LANGUAGE MODEL ELLIPSES ARE SIGNATURES

The model ellipse can be interpreted as a signature that associates outputs to the language model that produced them. To verify that a model produced an output, one can check its distance to the model's ellipse. If it is on the ellipse, it likely came from the model; otherwise, it almost certainly did not, since both in theory and practice the likelihood of an output falling on the intersection of two ellipses is extremely low. To measure the distance of a logprob $\boldsymbol{\ell}$ to the ellipse, we inspect the magnitude of the ellipse's inverse affine transform applied to the logprob, i.e., $(\boldsymbol{W}^+\boldsymbol{C}^+\boldsymbol{C}\boldsymbol{\ell} - \boldsymbol{\beta})/\boldsymbol{\gamma}$, where $\cdot^+$ denotes a

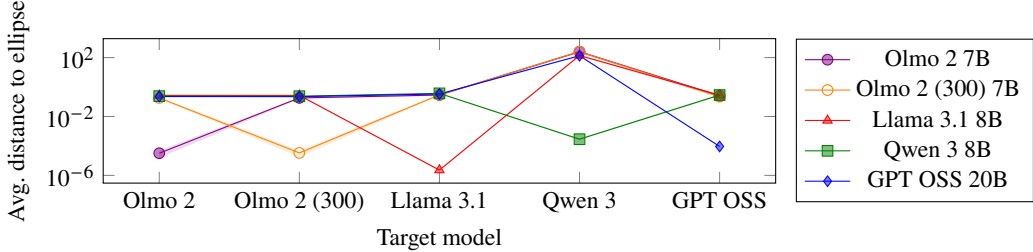

Figure 3: Mean distance to the model ellipse for logprobs generated from several open-weight models, projected onto each other's output spaces. Small distances indicate that the outputs are on the target model ellipse, and therefore came from the target model. Standard errors, shown as shaded regions, are mostly too narrow to see.

pseudoinverse. If the logprob came from the model, then this transformation should map the logprob back onto the unit sphere. Thus, we can interpret the deviation of the magnitude from 1 as a distance from the ellipse (in a linearly transformed space).[4]

To evaluate the effectiveness of language model signatures at identifying language model outputs, we generate a set of logprob outputs from four popular open-weight language models: Olmo 2 7B (OLMo et al., 2025), Llama 3.1 (Grattafiori et al., 2024), Qwen 3 8B (Yang et al., 2025), and GPT OSS (OpenAI et al., 2025). Since most of these models have different vocabularies and column spaces, we find a set of tokens that are common to all language models, then map each model's outputs onto the column space of each other model such that the cross-entropy between the original and projected outputs is minimized for the shared tokens. This essentially copies the linear signature of the target model onto the logprobs generated by another model. Finally, we apply the inverse affine transform and check the distance of the logprobs to the unit sphere for each model. Plotting these distances in Figure 3, we indeed find that the generating model always has the smallest distance to the unit sphere by several orders of magnitude. We also include the second-to-last Olmo 2 checkpoint (300) to compare with Olmo 2. We find that even in this case, the ellipse signature cleanly identifies the generating model.

As mentioned in Section 1, ellipse signatures have several unique properties that set them apart from existing output verification systems. In particular, ellipse signatures are *naturally occurring* because all modern language models have a final normalization layer; they are *self-contained* because the signature-checking procedure does not depend on the model input or parameters (except $W$, $\gamma$, and $\beta$); and they are *compact and redundant* because every single logprob output bears the ellipse signature.

### 2.3 COMPARISON TO EXISTING METHODS

To understand the niche that ellipse signatures fill, it is worthwhile to consider the breadth of existing methods for identifying models by their outputs, and how they differ from ellipse signatures. Many of these methods fall under the umbrella of language model fingerprinting, though we do not consider the ellipse signature itself to be a fingerprint because it lacks certain properties, namely robustness and stealthiness (Xu et al., 2025b, Section 2.3). Note that differences in properties do not make one method "better" than another—we highlight differences to show model ellipses and the compared methods are useful in different scenarios. Various methods and their properties are summarized in Table 2 in Section B. We are unaware of any existing model identification systems that exhibit all of the ellipse signature properties simultaneously.

Text-based watermarks (Liu et al., 2024), a subclass of fingerprint methods, inject signals into the text generation process that can later be used to identify the generating model. This is usually accomplished via sampling strategies (e.g., Kirchenbauer et al., 2023; Christ et al., 2024; Hou et al., 2024) which are sometimes distilled back into the model (Xu et al., 2025a, e.g.,). Watermarks are not naturally occurring, since they require intentional implementation on the model provider side. They are also not compact, since they require gathering statistical evidence over many generation steps.

---

[4]Section H discusses of how logit transformations like temperature affect this result.

One common approach to fingerprinting is to train backdoors into language models so that the model responds to specific inputs in a way that reveals its identity (e.g., Li et al., 2022). These are neither naturally occurring, nor self-contained, since they require special training, and require control over the model input in order to elicit the identifying response.

Some methods use naturally occurring fingerprints in output text to identify the generating language model. These include analysis of patterns in chain-of-thought outputs (Ren et al., 2025), as well as more straightforward training of classifiers for the task (Mitchell et al., 2023). Though closer in nature to ellipse signatures due to their natural occurrence, these are not compact because they require multiple generation steps to identify the generating model.

Sun et al. (2024) propose a zero-knowledge proof for language models (zkLLM), which can guarantee that a language model produced an output without sharing any model details, making it self-contained and hard to forge (with strong guarantees). While this method gives much stronger guarantees about how an output was produced than a model ellipse, it comes with the drawback that it makes inference much more expensive. Naturally occurring model ellipses on the other hand, do not.

The most similar method to ours is the method proposed by Finlayson et al. (2024) and expanded upon by Yang & Wu (2024), which uses a linear signature to identify models. The main difference between ellipse signatures and linear signatures is that only linear signatures are easy to forge while ellipse signatures are hard, as we will show in the next section.

## 3 ELLIPSE FORGERY IS HARD

In the context of model signatures, forgery means generating new outputs that conform to the model constraints without direct access to the model parameters. Formally, given a set of model outputs $x_1, \ldots, x_n$ and a black box signature verifying function $f$ such that $f(x_i) = 1$ for all $x_i$, forgery requires producing a new output $\hat{x}$ which passes verification, i.e., $f(\hat{x}) = 1$. In the case of the ellipse signature, $f(x) = 1$ if and only if $x$ is on the model ellipse.

Linear signatures (Finlayson et al., 2024; Yang & Wu, 2024) are easy to forge, due to results by Finlayson et al. (2024); Carlini et al. (2024). Their methods can forge a linear signature by extracting the linear constraints from an API, then producing new logprobs that satisfy those linear constraints (as we did in §2). In this section, we will demonstrate that, while it is *possible* to forge an ellipse signature by extracting the ellipse constraint from an API, it is practically infeasible for sufficiently large models using current known methods. Ellipse signature forgery resistance therefore relies on the idea that, as far as we are aware, there is no known method to produce new points on an ellipse without first fitting an ellipse to the known points. Here, we will make the same API assumptions used in Carlini et al. (2024); Finlayson et al. (2024); Nazir et al. (2025); Morris et al. (2024), where the API allows users to specify a prompt and then returns logprobs for a fixed set of at least $d$ tokens at every generation step. These assumptions are reasonable, given that OpenAI's Completions and Chat Completions APIs meet the criteria, specifically because the `logit_bias` parameter allows users to find the logprob of any token, as shown in Morris et al. (2024).

Carlini et al. (2024) present a method for extracting the ellipse from model outputs. The general idea is to use a fitting algorithm to find the ellipse that these outputs lie on. The parameters of the ellipse of best fit correspond to the singular values, rotation, and bias values in the model's final layer. We summarize the procedure in Algorithm 1 and present a more comprehensive overview in Section C. We then show that ellipse extraction is hard for at least two reasons. First, the query complexity of extracting enough logprobs from an API to determine the ellipse is $O(d^3 \log_v d)$, and second, the time complexity of the algorithm for fitting an ellipse to the outputs is $O(d^6)$.

### 3.1 ELLIPSE FITTING ERRORS FROM SMOOTHING

In theory, the $\varepsilon$ term in the denominator of a normalizatoin layer can cause issues for recovering the model parameters. Instead of enforcing $\|\text{norm}(x)\|_2 = 1$, these layers enforce $\|\text{norm}(x)\|_2 < 1$, meaning that outputs fall within the *interior* of the ellipsoid, instead of on the *surface*. This effect decreases for larger models (see Section G), and in practice, can cause ellipse fitting to fail.

In our own experiments, we found that the SVD-based ellipse fitting from Carlini et al. (2024) would sometimes fail for smaller models because the parameter $E$ output by the fitting algorithm was not

---

**Algorithm 1** Get output layer parameters of a language model.

**function** GET PARAMETERS( logprobs $\boldsymbol{\ell}_1, \ldots, \boldsymbol{\ell}_n \in \mathbb{R}^v$ )
    $\boldsymbol{WH} = \begin{bmatrix} \boldsymbol{\ell}_1 & \cdots & \boldsymbol{\ell}_n \end{bmatrix} \in \mathbb{R}^{v \times n}$     ▷ Create matrix from logprob outputs
    $d = \text{rank}(\boldsymbol{CWH})$     ▷ Find embedding size of model
    $\boldsymbol{A} = \boldsymbol{I}_{1:d}$     ▷ Choose a down-projection
    $\boldsymbol{A}^- = \boldsymbol{CWH}(\boldsymbol{AC}(\boldsymbol{WH})_{1:d})^{-1}$     ▷ Solve for up-projection
    $\boldsymbol{E}, \boldsymbol{b} = \text{ELLIPSOIDFIT}(\boldsymbol{AC}\boldsymbol{\ell}_1, \ldots, \boldsymbol{AC}\boldsymbol{\ell}_n)$     ▷ Solve for ellipse
    $\boldsymbol{U}, \boldsymbol{\Sigma}, \_ = \text{svd}(\text{Cholesky}(\boldsymbol{E}^{-1}))$     ▷ Convert ellipse to affine form
    **return** $\boldsymbol{A}^-, \boldsymbol{\Sigma}, \boldsymbol{U}, \boldsymbol{b}$     ▷ Return up-projection, stretch, rotation, and bias.
**end function**

---

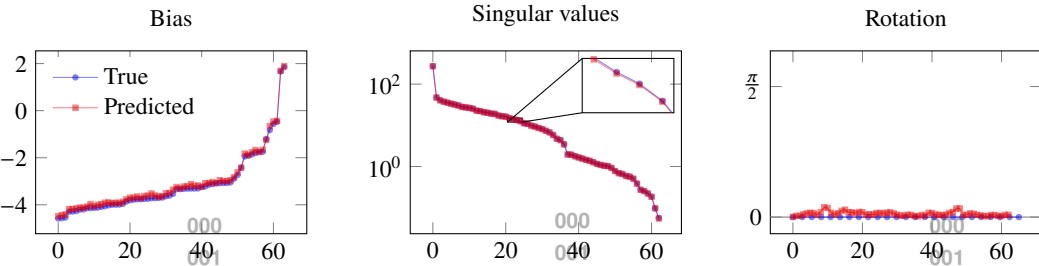

Figure 4: The predicted and true values for bias and singular values, and the angles between columns of the predicted and true rotation matrices for a 1 million-parameter model. The angles must lie in the range $[0, \pi]$, and the true rotation matrix columns have angle 0 with themselves. Our predictions are highly accurate, demonstrating robustness to normalization smoothing in ellipse fitting algorithms.

positive definite. The failure stems from the fact that SVD-based fitting is not *ellipse-specific*. The algorithm fits a *quadric* surface to the points, which may or may not be an ellipse, especially in the presence of noise from the $\varepsilon$ term. To guarantee the validity of our recovered ellipse, we turn to ellipse-specific fitting algorithms. We use fast algorithms for multidimensional ellipse fitting using semidefinite programming (Calafiore, 2002; Ying et al., 2012), in particular the ellipse fitting method from Ying et al. (2012), which is straightforward to implement, fast, and stable. We further detail our implementation in Section F. In general, we find that smoothing does not hurt ellipse fitting for models of sufficient size, confirming an observation from Carlini et al. (2024).

We test our ellipse-specific method on a small, pretrained, open-source model. In particular, we use a 1-million-parameter language model with an embedding size of 64 and a final layer norm with $\varepsilon = 1 \times 10^{-5}$ (Black et al., 2022; Eldan & Li, 2023). We obtain outputs from the model using the Pile dataset (Gao et al., 2020), then use the outputs to fit an ellipse and compare the predicted and true parameters. Figure 4 visually demonstrates their high similarity. As expected, we observe a consistent, slight underestimation of the model's singular values due to $\varepsilon$ smoothing.

To measure the benefits of using more outputs to fit the ellipse, we quantify the similarity between the estimated and true rotation, stretch, and bias for varying numbers of outputs in Figure 5. We measure bias and stretch similarity using mean squared error (MSE). For the rotation similarity between the predicted rotation $\boldsymbol{U}$ and true rotation $\boldsymbol{U}^*$, we use the geodesic distance $\text{tr}(\boldsymbol{U}^\top \boldsymbol{U}^*)$. Overall, we find that using more output samples generally improves parameter predictions, but has diminishing returns as irreducible error due to $\varepsilon$ begins to dominate.

## 3.2 THE SAMPLE COST OF MODEL ELLIPSE RECOVERY IS SUPER-CUBIC

Recovering the ellipse of an LM requires $O(d^2)$ outputs. An ellipse is a special case of a quadric surface (or simply quadric), which has the general equation

$$\sum_{i=1}^{d} \sum_{j=i}^{d} Q_{i,j} x_i x_j + \sum_{i=1}^{d} P_i x_i = 1, \tag{2}$$

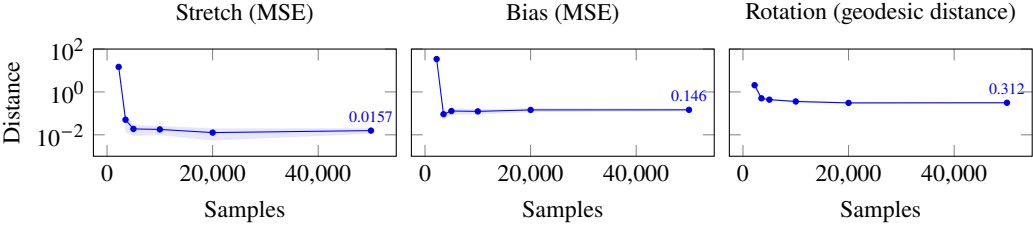

Figure 5: Distance between predicted and true parameters for bias, stretch, and rotation (lower is better). Fitting the ellipsoid to more samples generally improves the predictions, though with diminishing returns.

| Model | Hidden size $d$ | Vocab size | Samples for Ellipsoid | Ellipse ($) |
|---|---|---|---|---|
| `pythia-70m` | 512 | 50,304 | 131,327 | |
| `babbage-002` | 1536[a] | 101,281 | 1,180,415 | 1056 |
| `gpt-3.5-turbo` | 4650[b] | 101,281 | 10,813,574 | 150,699 |
| `llama-3-70b-instruct` | 8192 | 128,256 | 33,558,527 | 16,487,421[c] |

[a] Confirmed size from Carlini et al. (2024).
[b] Estimated size upper limit from Finlayson et al. (2024).
[c] Hypothetical cost based on inference cost for `gpt-4-turbo`.

Table 1: Models, their sizes, the number of samples required to ascertain their output ellipse, and the cost of the samples, based on OpenAI inference pricing on September 16, 2025. The number of samples required grows quadratically with the hidden size of the model, and the price grows cubically.

where the symmetric matrix $Q \in \mathbb{R}^{d \times d}$ and $P \in \mathbb{R}^d$ are parameters. An ellipse has the property that $Q$ is positive definite. The set of vectors $x \in \mathbb{R}^d$ that satisfy this equation form the ellipse surface. Since the total number of terms in the above equation is $d(d + 3)/2$, and the equation for a quadric is linear in its parameters, a set of $O(d^2)$ points is required in the general case to uniquely define an ellipse. In fact, in the worst case $\Omega(d^2)$ samples are required to find even a single new (not in the set of samples) point on the ellipse, since if the samples are in general position then for every point not in the samples there is an ellipse that includes the samples but not the point.

For a model with an RMS-norm and a modest embedding size $d = 2^9 = 512$, we would need at least $2^{17} + 3 \cdot 2^8 = 131,840$ outputs. For Llama 3 8B, the embedding size is $2^{12} = 4096$, and we would need 8,394,752 outputs. This quadratic growth makes finding a model's ellipse from its outputs much more expensive than extracting the model's column space, which only requires $O(d)$ samples.

In order to minimize cost, an attacker typically sends a single prefix token to an LM API for each sample. However, as the required number of samples surpasses the vocabulary size of the model it becomes necessary to send multi-token prefixes to the model in order to expand the number of unique prefixes. The number of tokens per sample grows logarithmically with the number of samples required, or $O(\log d)$. This would bring the overall API cost to $O(d^2 \log d)$ queries.

If the API only reveals a constant number of token logprobs per query, as has been the case historically for many API providers, the attacker needs to send multiple queries to recover the full logprob vector. Here, the attacker can save on cost by collecting $d$ full logprobs, then only collecting logprobs for the same subset of $d$ tokens for subsequent outputs, solving for the missing logprobs numerically later on. Even so, the attacker will still make $O(d)$ queries per sample. In all, this means that the cost of discovering the model ellipse grows at a rate of $O(vd + d^3 \log d)$, where the $vd$ term accounts for the cost of collecting the initial $d$ full logprobs.

Since the cost grows super-cubically with the embedding size of the model, current API pricing makes it prohibitively expensive to obtain the ellipse of many popular LMs, as shown in Table 1. Though OpenAI's cheapest and smallest available generative model, `babbage-002`, would cost about $1000 to attack, `gpt-3.5-turbo` would cost over $150,000. A 70B-scale model with inference cost similar to `gpt-4-turbo` would cost over $16 million.

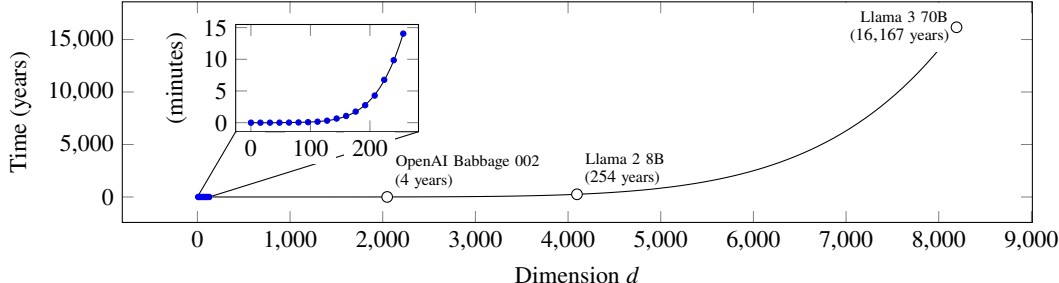

Figure 6: Extrapolated running time of our implementation of the ellipse extraction algorithm from Ying et al. (2012). We extract ellipses from several models with different hidden sizes (blue). Fitting a degree-6 polynomial to these points, we can extrapolate to guess at the time required to extract the parameters of larger, more popular models.

### 3.3 ELLIPSOID FITTING TAKES SEXTIC TIME

Obtaining sufficient samples from an API-protected language model is only the first step in finding the model ellipse. It turns out that the second step, fitting an ellipse to the samples, is prohibitively expensive computationally. Both the SVD-based and ellipse-specific fitting methods discussed in the previous section typically require $O(d^6)$ time, the time required to solve $O(d^2)$ equations of $O(d^2)$ variables (Carlini et al., 2024). Faster methods, such as those based on Strassen's algorithm ($\approx O(n^{2.807})$), are possible in the non-ellipse-specific case, but the absolute lower bound for such speedups is $O(n^2)$ (whether this bound is tight is an open question in computer science), which still leaves us with a complexity at least $O(d^4)$. The best known ellipse-specific method still requires $O(d^6)$ time and $O(d^4)$ space (Lin & Huang, 2016).

To evaluate the real-world practicality of ellipse fitting, we implement the algorithm from Lin & Huang (2016), and record the solving times for ellipses in dimensions ranging from 8 to 256. The times in Figure 6 are obtained by running the algorithm using 64 CPUs.[5] Extrapolating the best-fit polynomial of degree 6, recovering parameters from a typical 70 billion parameter model would take thousands of years.[6]

We leave it as an open question as to whether there exists a fast algorithm for generating new outputs on an unknown model ellipse based on samples. We ourselves tried a handful of optimizations to speed up ellipse fitting, such as parallelization on GPUs and approximation. Unfortunately, the memory requirements of ellipse fitting makes GPU acceleration infeasible for sufficiently large models, and approximation methods catastrophically degraded the accuracy of our recovered ellipse parameters.

## 4 ELLIPSE SIGNATURES ARE MESSAGE AUTHENTICATION CODES

The hardness of extracting ellipses, combined with the cheapness of checking logprobs against the ellipse, creates a type of trapdoor function which can be used to verify model outputs. In particular, we can interpret the model ellipse as a secret key, and the logprob outputs as a message, in analogy to keys and messages in cryptographic message authentication systems (Pass & Shelat, 2010).

In such systems, a *signer* sends a message and a *message authentication code* (MAC) to a *verifier*. The verifier uses the MAC to confirm the authenticity of the message. This process works as follows: first, the signer shares a secret key with the verifier. To send a verifiable message, the signer generates a tag from the key and message, e.g., by hashing their concatenation, and sends the message and tag to the verifier. The verifier can confirm the authenticity of the message by replicating the tag with their own copy of the secret key, e.g., by again hashing the concatenation of the key and message.

In the case of the model ellipse, the ellipse typically is chosen via the model training, but could also be chosen by sampling a random ellipse. Signing occurs when the model generates logprobs. The

---

[5]We run our experiments on AMD EPYC 7643 48-core processors.

[6]Such extrapolations should be taken with a grain of salt—these numbers are intended only to give intuition.

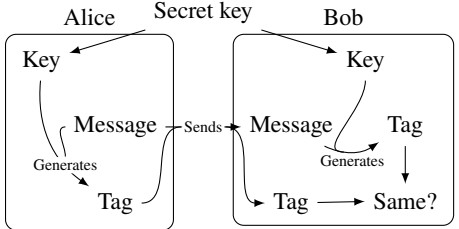 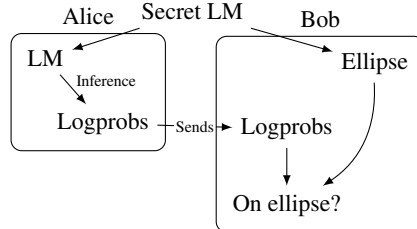

Figure 7: A cryptographic message authentication system (left) uses a shared secret key to validate messages. The message author, Alice, signs the message by generating a tag from the message and a secret key. After receiving the message and tag, Bob verifies the message by generating another tag with his copy of the key. If the received and generated tags match, the message is authentic. In our proposed verification system (right), the ellipse functions as a secret key, and logprobs take the role of messages. The tag is encoded in the position of the logprobs in $R^v$, and can be verified by checking that they lie on the secret ellipse.

logprobs contain both the message (information about the next-token distribution) and the tag, which is encoded in the logprob's position in $\mathbb{R}^v$. To verify the message, someone with the secret model ellipse can verify that the logprob's position is on the ellipse.

The security of a MAC hinges on the infeasibility of forgery. For a MAC, forgery means producing message-tag pairs that pass verification (and have not previously been generated) without access to the secret key. For the ellipse signature, a would-be forger's task is to efficiently produce a logprob that lies on the model ellipse. The most obvious forgery attack would be to steal the model ellipse by collecting outputs, solving for the ellipse, then using the ellipse to produce new logprobs. As discussed in Section 3, this is impractically hard for production-size language models of today.

Because ellipse verification guarantees only that individual logprob vectors came from a specific model, it is possible that an attacker could create a plausible token sequence $g$ of length $n$ by saving a collection of logprob outputs from a target model, then piecing them together in a sequence $\ell^{(1)} \cdots \ell^{(n)}$ such that the top token from each logprob corresponds to a token in $g$, i.e., $\arg\max_i \ell_j^{(i)} = g_j$. There are two potential defenses against this type of attack. The first would be to give the verifier access to a database of all outputs ever produced by the provider. At scale, this might be prohibitively costly to maintain. A second deterrent to this kind of attack is that logprobs have been shown to contain a large amount of information about their prefix (Morris et al., 2024; Nazir et al., 2025), meaning it is possible to train *language model inverters* that reliably reconstruct the prefix of all logprob outputs. If the verifier finds that the inverter assigns low likelihood to the first half of the sequence when conditioned on the second half, then it is likely that the sequence has been tampered with.

A message authentication protocol for language models can be a tool for language model accountability. As a hypothetical scenario, suppose that language model providers are required by law to share their ellipse with a trusted third party. Then, if a user receives a harmful output from the model and sues for damages and the provider denies generating the output, the third party can provide convincing evidence about which party is correct due to the forgery-resistant property of language model ellipses.

## 5 DISCUSSION AND CONCLUSION

Because of their unique properties, ellipse signatures fill a new niche in the space of language model verification methods. This combination of their features is especially promising for doing language model forensics, since ellipse signatures can be used to identify any model with high certainty, even if the output provider did not intend to sign their outputs.

Ellipse signatures for language models can be improved on at least three fronts. First, the hardness of forging ellipses is only polynomial, far from a cryptographic security guarantee. It is likely possible to identify other constraints on model outputs that give stronger guarantees. Second, our proposed ellipse signature protocol requires that the API provides logprobs. As of this writing, OpenAI is the only major commercial provider that allows access to logprobs, and even this access is limited to a handful of models through ad hoc querying workarounds. Lastly, the ellipse signature does not

have the desirable property of being difficult to remove (often a goal for model fingerprints), since modifying the model outputs or parameters erases the signature by breaking or changing the ellipse constraints. Future work could explore other kinds of signatures for language models, which are hard to remove. We hope that ellipse signatures will offer an exciting new avenue for research into such signatures, which will greatly impact language model security, accountability, and forensics.

## 6 ACKNOWLEDGEMENTS

Matthew Finlayson is supported by a fellowship from the National Science Foundation (NSF) Graduate Research Fellowship Program. Swabha Swayamdipta's research is supported in part by the NSF under grant IIS2403437, the Simons Foundation, Apple, Intel and the Allen Institute for AI. Part of this research was done when Swabha Swayamdipta and Matthew Finlayson were visitors at the Simon's Institute. Xiang Ren's research is supported in part by the Office of the Director of National Intelligence (ODNI), Intelligence Advanced Research Projects Activity (IARPA), via the HIATUS Program contract #2022-22072200006, the Defense Advanced Research Projects Agency with award HR00112220046, and NSF IIS 2048211. We would like to thank all the collaborators in INK and DILL research labs at USC for their constructive feedback on the work.

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

| Method | Examples | Naturally occurring | Self contained | Compact | Forgery resistant |
|---|---|---|---|---|---|
| Text-based watermark | (Kirchenbauer et al., 2023; Hou et al., 2024) | No | Yes | No | Yes |
| Backdoor fingerprint | (Li et al., 2022) | No | No | - | - |
| Natural fingerprint | (Ren et al., 2025) | Yes | - | No | - |
| Cryptographic | (Sun et al., 2024) | No | Yes | Yes | Yes |
| Linear signature | (Finlayson et al., 2024; Yang & Wu, 2024) | Yes | Yes | Yes | No |
| Ellipse signature | | Yes | Yes | Yes | Yes |

Table 2: Summary of different model attribution methods and their properties. Some method classes may include both methods with and methods without specific properties, in which case we assign the label "-".

## A   CENTERED LOGITS ALSO LIE ON AN ELLIPSE

The *centering* operation on a vector subtracts the mean value of the vector entries, so that the sum of the entries is 0. Centering a vector $x \in \mathbb{R}^d$ can be computed as $x - \sum_{i=1}^{d} x_i / d$, which is sometimes denoted as $x - \mathbb{E}[x]$. Centering is a linear operation, since $x - \mathbb{E}[x] = (I - \frac{1}{d})x$. We will call the matrix associated with this operation $C = I - \frac{1}{d}$. The softmax function $\text{softmax}(x) = \exp x / \sum_{i=1}^{d} \exp x_i$ has a special relationship with centering. First, since the softmax function is invariant to scalar addition, i.e., $\text{softmax}(x) = \text{softmax}(x + c)$, it follows (by substituting $\mathbb{E}[x]$ for $c$) that it is also invariant to centering, i.e., $\text{softmax}(x) = \text{softmax}(Cx)$. Secondly, centering the log of a softmax is equivalent to centering, i.e., $C \log \text{softmax}(x) = Cx$.

This property is useful because language model APIs typically only give log-probabilities (logprobs), not logits. While we cannot recover the logits from logprobs directly, centering the logprobs gives us the same result as centering the logits. Thus, when observing outputs from an LM API, we may have access only to centered logits $CW(\gamma \odot \hat{x} + \beta)$. Luckily, because $C$ is linear, these outputs also lie on the surface of an ellipse.

## B   COMPARISON TO EXISTING VERIFICATION SYSTEMS

Language model watermark methods (Liu et al., 2024) leave telltale signs in model-generated text that can be detected later on. For instance, Kirchenbauer et al. (2023) restrict language model sampling to specific subsets of the vocabulary at each generation step using a deterministic rule, and use hypothesis testing to verify that text has been watermarked. Christ et al. (2024) develop a variant of this type of method that is *undetectable* except to those who have a secret key. These methods require that the model provider implement specific decoding schemes at generation time, and require a sufficiently long generation to accumulate evidence that a text has been watermarked. In contrast, ellipse-based verification requires no changes on the provider side, and each generation step of the model independently bears the model's signature, meaning that a generation of length 1 is sufficient to identify the source model.

Similarly, since our system is analogous to a MAC system, one might ask if the method would not be improved by using a proper MAC system with hard cryptographic guarantees. Indeed this is the case, except for the fact that, once again, implementing a MAC system requires changes on the API provider side, whereas ellipse-based verification does not.

Fingerprints for language models are a diverse set of methods, usually considered distinct from watermarks, because they are generally incorporated into the model weights themselves during training, and are designed to be hard to detect and hard to erase. The most similar fingerprinting methods to our ellipse-based verification are those that identify naturally occurring fingerprints (Zeng

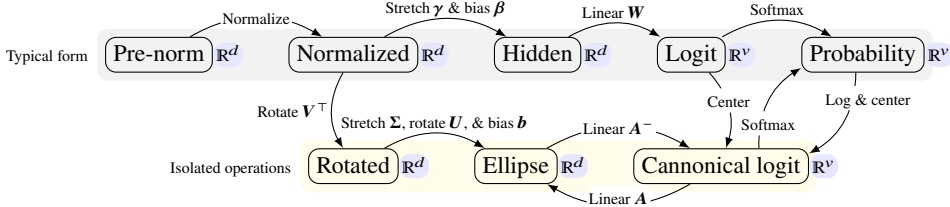

Figure 8: An overview of the intermediate representations in both traditional (gray) and our own (yellow) parameterization of the final LM layers. Representations (boxes) are annotated the functions that map between them (arrows), as well as the space in which they reside (blue tags). Our method recovers the parameters $\Sigma$, $U$, $b$, and $A^-$, which in turn give us access to the *cannonical logit*, *ellipse*, and *rotated* representations. Notably, the *rotated* representation that we recover is a pure rotation of the normalized representation.

et al., 2025, e.g.,), including one based on the proposed model identification method from Finlayson et al. (2024) which uses the uniqueness of the linear subspace spanned by the columns of the softmax matrix (Yang & Wu, 2024). The main difference between our method and the latter is somewhat subtle: in the Yang & Wu (2024) method, it is be easy to copy the fingerprint from one model onto another, even when access is limited to the API, allowing one model to pose as another. Our ellipse method guarantees that it is computationally difficult to "steal" the ellipse of an API model.

Finally, one might consider an alternative input verification method, similar to ours, where the verifier has access to the model. The verifier can then check that outputs from the API matches the output from their own model for any specific input. The main drawback of this method is that the verifier needs access to the whole model, as well as the input. One can imagine a situation where the model provider does not wish to reveal all model parameters, or has a proprietary system message that they do not wish to share (though best practice dictates that prompts should not be considered secrets (Zhang et al., 2024; Morris et al., 2024; Nazir et al., 2025)). In these cases, it may be preferable to use an ellipse-based validation scheme, since it only requires revealing parameters from the final layers.

## C    How to extract an ellipse

Here we will show how to recover the ellipse parameters from model outputs. The general idea will be to collect model outputs, then use a fitting algorithm to find the ellipse that the outputs lie on. The parameters of the ellipse of best fit will correspond to the model parameters. The procedure described in this section is summarized in Algorithm 1.

In order to recover the rotations, scales, and bias in the model's output layer, we will reformulate the model in a way that exposes these operations. We illustrate this reparameterization in Figure 8. Typically, the final layers have the form

$$x \mapsto \mathrm{softmax}(W(\gamma \odot \hat{x} + \beta)), \tag{3}$$

with an unembedding matrix $W$ and element-wise affine transform $\gamma, \beta \in \mathbb{R}^d$ applied to the normalized input $\hat{x}$. We can reparameterize the argument to the softmax with an equivalent affine transformation of $\hat{x}$

$$x \mapsto \mathrm{softmax}(A^-(U\Sigma V^\top \hat{x} + b)), \tag{4}$$

where $A^- \in \mathbb{R}^{v \times d}$ is a matrix, $U, V^\top \in \mathbb{R}^{d \times d}$ are unitary (i.e., rotation) matrices, $\Sigma \in \mathbb{R}^{d \times d}$ is a diagonal (i.e., scaling) matrix, and $b \in \mathbb{R}^d$. We claim that our formulation in Equation (4) is equivalent to Equation (3) when, for a chosen full-rank matrix $A \in \mathbb{R}^{d \times v}$, we have that $U, \Sigma, V^\top$ is the singular value decomposition of $ACW \odot \gamma$, $b$ is $ACW\beta$, and $A^-$ is a generalized inverse of $A$, computed as $CW(ACW)^{-1}$ (proof in Section D).

Our re-parameterization is useful because it isolates distinct operations on the LM representations. The output is first rotated by $V^\top$, scaled along the axis directions by $\Sigma$, rotated again by $U$, translated by $b$, then projected into a higher dimensional space by $A^-$. In the original formulation, these operations all occur implicitly via $CW \odot \gamma$ and $CW\beta$.

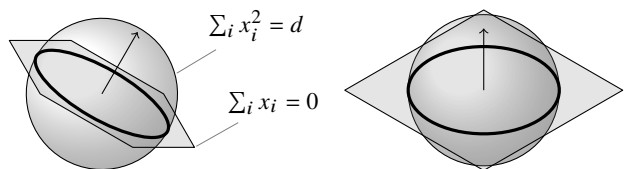

Figure 9: An illustration of the layer norm output space (left), which is shown as the thick ring at the intersection of the plane perpendicular to $\mathbf{1}$ (due to centering) and a sphere (due to normalization). To reduce the dimension of the representation, we can apply an isometric transform that rotates $\mathbf{1}$ to align with an axis (right), then drop that axis.

Because model outputs are on a $d$-dimensional ellipse in $v$ dimensional space, we will first project them back down to $d$ dimensions before fitting an ellipse to them. For simplicity's sake, we choose the down-projection matrix $A \in \mathbb{R}^{d \times v}$ to be the first $d$ rows of the identity matrix $I_{1,\ldots,d}$ because multiplying by this amounts to truncating a vector after $d$ entries.

Now we can solve for our first parameter $A^-$, the up-projection matrix that restores down-projected vectors to their full dimension. We begin by collecting $d$ logprob outputs from the model and centering them. These outputs come from some unknown hidden states $h_1, \ldots, h_d \in \mathbb{R}^d$, which we consider to be stacked into a matrix $H \in \mathbb{R}^{d \times d}$, meaning our collected outputs are $CWH$. We are looking for $A^-$ such that $A^- ACW = CW$. This can be solved by inverting the down-projected outputs $A^- = CWH(ACWH)^{-1} = CW(ACW)^{-1}$.

Next, we turn our attention to finding the remaining parameters $U$, $\Sigma$, and $b$, which we will do by collecting a sufficient number of outputs, projecting them onto $\mathbb{R}^d$, and fitting an ellipse to them. Ellipsoid fitting algorithms generally return parameters $E \in \mathbb{R}^{d \times d}$ and $b \in \mathbb{R}^d$, where $E$ is symmetric and positive-definite. The ellipse is the set of points that satisfy $(x - b)^\top E (x - b) = 0$. We can obtain our model parameters as $b = b$ and $E^{-1} = (U\Sigma)(U\Sigma)^\top$, using Cholesky and singular value decomposition to find the latter. The parameter $U$ obtained by this method is not unique, since rotating an ellipse $\theta$ degrees in one direction yields the same surface as rotating it $180 - \theta$ degrees in the opposite direction. Nevertheless, we make it unique by specifying an arbitrary constraint that the columns of $U$ all have a positive value in their first nonzero entry.

Thus we have a method to obtain all the parameters of our LM's output layers (except $V^\top$, which the ellipse is invariant to).

## D    PROOF OF EQUIVALENT REPARAMETERIZATION

*Proof.* Our goal is to prove that
$$\text{softmax}(W(\gamma \odot \hat{x} + \beta)) = \text{softmax}(A^- U\Sigma V^\top \hat{x} + b)$$
when $U, \Sigma, V^\top = \text{svd}(ACW \odot \gamma)$, $b = CW\beta$, and $A^- = CW(ACW)^{-1}$, where $A \in \mathbb{R}^{d \times v}$.

$$\text{softmax}(W(\gamma \odot \hat{x} + \beta)) \tag{5}$$
$$= \text{softmax}((W \odot \gamma)\hat{x} + W\beta) \qquad \text{Distribute } W \tag{6}$$
$$= \text{softmax}(C((W \odot \gamma)\hat{x} + W\beta)) \qquad \text{Softmax invariant} \tag{7}$$
$$= \text{softmax}((CW \odot \gamma)\hat{x} + CW\beta) \qquad \text{Distribute } C \tag{8}$$
$$= \text{softmax}((CW \odot \gamma)\hat{x} + b) \qquad \text{Substitute } b \tag{9}$$
$$= \text{softmax}\left((CW(ACW)^{-1}ACW \odot \gamma)\hat{x} + b\right) \qquad (ACW)^{-1}(ACW) = I \tag{10}$$
$$= \text{softmax}\left(A^- U\Sigma V^\top \hat{x} + b\right) \qquad \text{Substitutions} \tag{11}$$
$$\square$$

## E    PARAMETER RECOVERY FOR LAYER NORM MODELS

Up to this point, we have considered models with RMS-like norms, but the details of parameter recovery become slightly more involved when the model uses a layer norm function. The layer norm

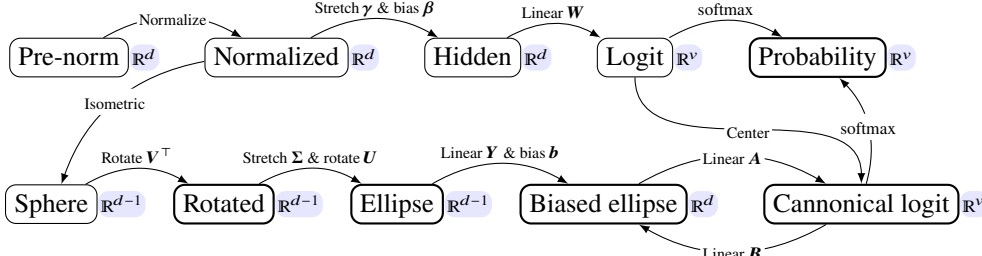

Figure 10: Model representations and mappings between them for LMs with a layer norm. Compared to Figure 8, this model has an ellipse in $d-1$ dimensions.

function consists of centering and normalization

$$\text{layernorm}(\boldsymbol{x}) = \frac{\boldsymbol{x} - \mathbb{E}[\boldsymbol{x}]}{\sqrt{\varepsilon + \text{Var}[\boldsymbol{x}]}} \, ,$$

where $\text{Var}[\boldsymbol{x}]$ is the variance of the entries of $\boldsymbol{x}$. In this setting, the ellipse lives in a $d-1$-dimensional space due to the fact that the layer norm centers the input before normalizing. The centering operation maps the input onto a plane by enforcing that $\sum_{i=1}^{d} x_i = 0$, and the normalization maps it onto the closest point on the sphere by enforcing that $\sum_{i=1}^{d} x_i^2 = d$. Figure 9 shows how the output space of the layer norm is a *two*-dimensional sphere (i.e., a circle) when $d=3$. A consequence of this is that the bias term in $\mathbb{R}^d$ now "lifts" the ellipse off the plane on which it resides into a $d$-dimensional space.

To account for this, we modify our reparameterization as shown in Figure 10. In particular, we first project the sphere into $\mathbb{R}^{d-1}$ to apply the linear ellipse transformation, then lift the ellipse into $\mathbb{R}^d$ to apply the bias term. Our projection onto $\mathbb{R}^{d-1}$ is an isometric linear transformation which first applies a rotation that maps the vector $\mathbf{1}$ to the vector $(\sqrt{d}, 0, 0, \ldots, 0)$ (as shown in Figure 9) and then drops the first entry, which will always be zero for any centered input. Because it is an isometric transformation, the projected representations are still on a sphere. When it comes time to apply the bias, we apply an up-projection $\boldsymbol{Y}$ before adding $\boldsymbol{b}$.

When recovering an unknown $\boldsymbol{b}$ from a set of layer norm logprobs, we modify our algorithm to account for our reformulation. The key idea is to subtract a point on the biased ellipse from every other point so that their new plane intersects the origin. We can then apply a down-projection as we have done before by simply dropping an axis. We find the matrix $\boldsymbol{Y}$ that inverts this axis dropping in the same way that we found $\boldsymbol{A}^-$.

---

**Algorithm 2** Get output layer parameters of a language model *with a layer norm*.

---

**function** GET PARAMETERS( logprobs $\boldsymbol{\ell}_1, \ldots, \boldsymbol{\ell}_n \in \mathbb{R}^v$ )
    $\boldsymbol{WH} = \begin{bmatrix} \boldsymbol{\ell}_1 & \cdots & \boldsymbol{\ell}_n \end{bmatrix} \in \mathbb{R}^{v \times n}$         ▷ Create matrix from logprob outputs
    $d = \text{rank}(\boldsymbol{CWH})$         ▷ Find embedding size of model
    $\boldsymbol{A} = \boldsymbol{I}_{1:d}^v$         ▷ Choose a down-projection
    $\boldsymbol{A}^- = \boldsymbol{CWH}(\boldsymbol{AC}(\boldsymbol{WH})_{1:d})^{-1}$         ▷ Solve for up-projection
    $\boldsymbol{b}_1 = \boldsymbol{AC}\boldsymbol{\ell}_1$
    $\boldsymbol{Z} = \boldsymbol{I}_{1:d-1}^d$
    $\boldsymbol{Y} = (\boldsymbol{ACWH} - \boldsymbol{b}_1)(\boldsymbol{Z}(\boldsymbol{AC}(\boldsymbol{WH})_{2:d} - \boldsymbol{b}_1))^{-1}$
    $\boldsymbol{E}, \boldsymbol{b}_2 = \text{ELLIPSOIDFIT}(\boldsymbol{Z}(\boldsymbol{AC}\boldsymbol{\ell}_1 - \boldsymbol{b}_1), \ldots, \boldsymbol{Z}(\boldsymbol{AC}\boldsymbol{\ell}_n - \boldsymbol{b}_1))$     ▷ Solve for ellipse
    $\boldsymbol{b} = \boldsymbol{b}_1 + \boldsymbol{Y}\boldsymbol{b}_2$     ▷ Solve for $\boldsymbol{b}$
    $\boldsymbol{U}, \boldsymbol{\Sigma}, \_ = \text{svd}(\text{Cholesky}(\boldsymbol{E}^{-1}))$     ▷ Convert ellipse to affine form
    **return** $\boldsymbol{A}^-, \boldsymbol{Y}, \boldsymbol{\Sigma}, \boldsymbol{U}, \boldsymbol{b}$
**end function**

---

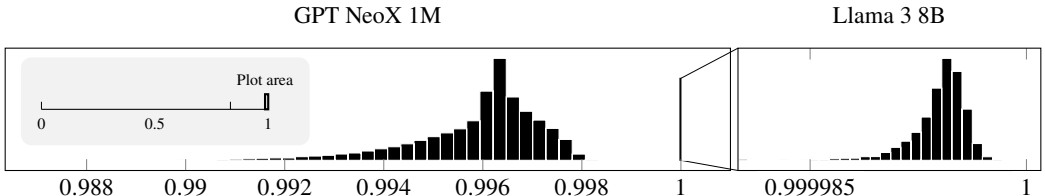

Figure 11: The distribution of L2 norms of hidden states $\hat{x}$ from two pre-trained language models with hidden sizes 64 (GPT NeoX) and 4096 (Llama 3). Both distributions are concentrated near the maximum norm of 1, more so for the larger model.

## F  ELLIPSE EXTRACTION IMPLEMENTATION

Our implementation is written with CVXPY and uses the MOSEK solver. To verify the correctness of our method, we first check that it solves for the exact parameters of a randomly initialized model with un-smoothed normalization (i.e., without a $\varepsilon$ term in the normalization denominator, as in Equation (1)). In this setting, our method shows negligible errors ($< 1 \times 10^{-15}$) in recovering the model parameters. We attribute these small errors to precision limitations.

## G  SMOOTHING SEVERITY

To get a sense of the severity of fitting problems due to smoothing, we plot a histogram of the norms of the normalized outputs for pre-trained models with $\varepsilon = 1 \times 10^{-5}$, shown in Figure 11. Fortunately, we find that the norms are generally clustered tightly around a point near 1. Notably, for larger models, this clustering becomes even tighter and closer to 1, likely because the smoothing term has less effect. Thus, in accordance with Carlini et al. (2024), we find that our method works fairly well if we simply *ignore* this potential source of error, so long as our fitting method is ellipse-specific and we oversample points from the model, or when the model is large enough in size. By ignoring $\varepsilon$, we expect that our recovered scaling terms will be slight underestimates.

## H  LOGIT TRANSFORMATIONS

Language model APIs may sometimes apply transformations to logits or model parameters which may affect the signature.

Top-$k$ truncation for $k < d$ ($k > d$ has no effect on the ellipse) means that outputs will still fall within (though no longer on) the down-projected signature in that specific top-$k$ direction. This hurts the usefulness of the signature, but if you see an output outside the ellipse projection you know the output came from a different model.

Temperature, i.e., scaling the logits before applying the softmax, so long as it remains constant, has the effect that the ellipse will be scaled, meaning that the outputs will not be on the ellipse, but they will all be exactly the same distance from it, allowing a verifier to still detect the signature.

Quantization/mixed precision, an efficient inference strategy where some model weights are converted to lower precision, should have no effect on ellipse signatures since the language model head is generally not quantized.

