# OpenReview forum: "Every Language Model Has a Forgery-Resistant Signature"
_ICLR.cc/2026/Conference — ICLR 2026 Oral_

### Official Review · Reviewer_pMYU · 2025-10-25

**Soundness:** 3
**Presentation:** 3
**Contribution:** 3
**Rating:** 6
**Confidence:** 4

**Summary:**

This paper proposes a novel method for verifying the output attribution of closed-weight LMs, termed the "Ellipse Signature." The authors build upon a key geometric observation: due to the normalization layer (mapping hidden states to a sphere) and the subsequent linear layer (an affine transform) at the end of modern LM architectures, the model's outputs (logits) are strictly constrained to lie on the surface of a high-dimensional hyperellipsoid.The paper's core contribution is demonstrating that this "naturally occurring" signature is highly "forgery-resistant". In sharp contrast to prior work on "linear signatures", which are easy to forge (requiring only $O(d)$ queries), the authors theoretically and experimentally argue that "extracting" the ellipse signature is computationally prohibitive. It requires $O(d^2)$ API queries to acquire sufficient points and $O(d^6)$ computational time to fit the ellipsoid parameters. For a large model like Llama-3-70B, this is shown to be practically infeasible (estimated at over $16 million in cost and 16,167 years of compute). Finally, the authors frame this mechanism as a naturally occurring Message Authentication Code (MAC) system, where the ellipse parameters act as a "secret key," providing a powerful tool for model forensics and accountability.

**Strengths:**

1. The derivation of ellipsoid constraints from normalization geometry is direct and relies on realistic architectural assumptions.

2. Sample and computational complexity analysis shows that reconstructing the ellipsoid from API outputs is practically infeasible for large models.

3. Experiments demonstrate that outputs of the target model have significantly smaller distances to their own ellipsoid compared to other models, confirming discriminative power.

4. Framing the approach as a MAC-like verification mechanism helps conceptualize how providers might share or authenticate model signatures securely.

**Weaknesses:**

1. Real-world APIs typically expose only top-$k$ log probabilities or apply noise. The paper primarily uses full logits and small models. It should evaluate whether the detection remains reliable under top-k truncation, temperature sampling, quantization, mixed precision, or the $\varepsilon$-smoothing effects that shift points inside the ellipsoid. Although $\varepsilon$ and alternative fitting methods are discussed, they do not cover large models or diverse normalization types.

2. The analysis assumes static base models. In Model-as-a-Service (MaaS) settings, providers often fine-tune models (e.g., via SFT or LoRA). The signature is described as fragile, suggesting any parameter change may alter the ellipsoid. The paper does not analyze the sensitivity of the ellipsoid to such updates—whether a small LoRA change yields a distinct or slightly perturbed ellipsoid. This limits practical use since results only show separability between different base models, not versions or fine-tunes of the same model. The paper also omits discussion of robustness to quantization, distillation, or post-training modifications.

3. The paper does not seem to add the usage of large language models section, which is required by this year's submission requirements.

**Questions:**

See weaknesses

---

> ### Author Response · Authors · 2025-11-24
>
> Thank you for your positive evaluation and constructive comments. Please let us know whether the following addresses your concerns:
>
> ## Logprob availability
>
> You are correct that many APIs do not give logprobs, though notably OpenAI still exposes logprobs on some of their endpoints. Regardless, we believe that our findings remain relevant as model providers make decisions about what to expose in the future. It could be that more providers elect to expose logprobs if they wish to use ellipse-based model signatures.
>
> ## Topk-truncation, etc.,
>
> These are some interesting questions. We will address them here and in appendix H of the paper revision:
>
> Top-k truncation for k\<d (k\>d has no effect): outputs will still fall within (though no longer on) the down-projected signature in that specific top-k direction. This hurts the usefulness of the signature, but if you see an output outside the ellipse projection you know the output came from a different model.
>
> Temperature: so long as the temperature stays constant, the effect is that the ellipse will be scaled, meaning that the outputs will not be on the ellipse, but they will all be exactly the same distance from it, allowing the provider to still detect the signature.
>
> Quantization/mixed precision: since the language model head is generally not quantized, this should have no effect on the signature.
>
> Smoothing: section 3.1 goes into detail about the (non) effects of smoothing, and our experiments are all conducted on smoothed models. Appendix G explores the severity of errors from smoothing.
>
> Alternative norm types \- we cover the two most common normalization types, RMSNorm in the main paper and Layernorm in Appendix E.
>
> ## Static base models:
>
> We compare related models in Figure 3 where we show that the ellipses of the second-to-last and final checkpoints of Olmo2 have very distinct ellipses. It is possible that these ellipses are related in some way, and it probably does take some number of training steps for the ellipses to diverge. Nevertheless, robustness is not a realistic goal of the ellipse signature, at least in its current form, and fingerprints are likely a more appropriate tool when robustness is required.
>
> ## LLM usage statement
>
> The ICLR guidelines say
>
> > \[I\]f LLMs played a significant role in research ideation and/or writing to the extent that they could be regarded as a contributor, then authors should describe the precise role of the LLM in a separate section on LLM usage.
>
> Since LLMs did not play a significant role in our research process, we do not include this section. We declared this non-usage in one of the OpenReview fields next to our abstract

---

### Official Review · Reviewer_Cfrc · 2025-10-27

**Soundness:** 3
**Presentation:** 3
**Contribution:** 3
**Rating:** 6
**Confidence:** 3

**Summary:**

This paper uses the geometric constraint that language model outputs lie on a high-dimensional ellipse (ellipse signature) to identify source models. The signature is hard to forge, inherent to all models, self-contained, and compact. It evaluates ellipse extraction from small models, addresses scalability issues, and proposes an output verification protocol, advancing beyond methods like language model fingerprints.

**Strengths:**

The paper identifies and formalizes the ellipsoid constraint as a forensic signature, which is novel and interesting.

The derivation explaining why model outputs lie on high-dimensional ellipsoids—arising from the interaction between normalization layers and linear transformations—is theoretically sound and well-supported by comprehensive mathematical proofs, demonstrating strong theoretical innovation.

**Weaknesses:**

The paper lacks both quantitative and qualitative comparisons with existing black-box fingerprinting methods [R1, R2].

The O(d^6) fitting complexity makes the method computationally infeasible for modern large-scale models.

A key requirement for fingerprinting techniques is robustness, meaning that the fingerprint should remain invariant even after slight model modifications (e.g., LoRA fine-tuning). Evaluating the proposed ellipsoid-based approach under such conditions appears challenging for large models, though it may be feasible for smaller ones.

**Questions:**

None

---

> ### Comment · Reviewer_3e9D · 2025-11-14
>
> Isn't the fact that fitting requires complexity d^6 a positive for their paper? They are very clear that this is their argument for why it should be *unforgeable*! So it seems a bit odd to say "The O(d^6) fitting complexity makes the method computationally infeasible for modern large-scale models."
> Given the model parameters, they can extract the ellipse easily.

---

> > ### Comment · Reviewer_Cfrc · 2025-11-23
> >
> > Thanks for Reviewer 3e9D’s suggestions, and we apologize for our misunderstanding. This paper does not seem to explicitly declare that it is about LLM watermarking, and we think it is actually a white-box fingerprinting method for LLMs.

---

> ### Author Response · Authors · 2025-11-24
>
> Thank you for your positive evaluation and constructive feedback. Please let us know whether the following addresses your concerns:
>
> ## Comparisons
>
> In terms of qualitative comparisons, section 2.3 to contrasts our method with existing ones. In terms of quantitative comparisons, we reproduce here our response to Reviewer nUfp:
>
> > You bring up an important question about how to compare our method with existing methods. While writing the paper, we considered how best to show that the method works and explain each of the properties that sets the method apart from existing methods.
>
> > First, to show that the method works, we conducted the experiment in Figure 3, which shows that the target model ellipse is detectably distinct from other model outputs. Since this experiment showed that it is easy to detect which model generated an output using the ellipse, we did not think an accuracy comparison to other methods would be particularly fair or helpful, since our method already would achieve 100% accuracy. Several other methods achieve similarly high accuracy, e.g., under the right settings the watermark from Kirchenbauer (2023) gets perfect accuracy as well. Thus we do not think that accuracy is a very relevant comparison point between our method and others.
>
> > Next, we considered the properties that set our method apart, namely the “naturally occurring”, “self contained”, “compact and redundant” and “hard to forge” properties. Since these properties are binary (either a model has the property or it doesn’t) we couldn’t think of an experiment that would make sense here. For instance, it is not clear what experiment to run to measure whether a method “requires access to the model input”.
>
> > We believe that adding a summary comparison table for the different properties would be helpful for readers. We have included this as Table 2 in the appendix. Would this be sufficient to address your concern here? Are there particular experiments that you would like to see?
>
> > Lastly, we hope to convey that we are not claiming that ellipse signatures are superior to other methods (except for linear signatures, which only lack the hard-to-forge property), only that they are applicable under different scenarios and assumptions based on their properties. For this reason, we have added “we emphasize that signatures are not necessarily \\emph{better} than other methods, rather their properties make them more suitable in specific situations“ to the introduction.
>
> ## Feasibility
>
> Reviewer 3e9D is correct that the $O(d^6)$ complexity is a feature, rather than a drawback. It makes ellipse signature forgery hard. They are also correct that extracting the ellipse given access to the model parameters is not hard.
>
> ## Robustness
>
> We do not make any claims that our method exhibits robustness, a key feature of model fingerprints. For this reason, we do not view signatures as fingerprints, since they exhibit different properties that can be useful in different settings. You may find it interesting that in Figure 3 adjacent checkpoints of Olmo have completely different signatures (almost the opposite of robustness). We do not view this as a failure of signatures, since this fragility means that signatures are unique to their models, down to the level of individual checkpoints. We note that Yang & Wu (2024) are able to identify LoRA finetunes via the linear signature, and we are excited to see if future work can develop a similar method for ellipse signatures. We have revised our introduction to make it clearer in the introduction that our method is *different* rather than *better* than fingerprints and other methods (Paragraph 3)

---

> > ### Comment · Reviewer_Cfrc · 2025-11-27
> >
> > Could you provide an application scenario for the proposed method—for example, verifying the integrity of a model? I mean that if the signature lacks robustness, it may not be suitable for identifying a specific model.

---

> > > ### Author Response · Authors · 2025-11-27
> > >
> > > Great question!
> > >
> > > Here is an application scenario which does not rely on robustness: Alice claims that she has early API access to a not-yet-released language model. As a commitment to her claim, she publishes outputs from that model. Later on, when the model weights are released, anyone can verify her claims by checking that the outputs she published are on the ellipse.
> > >
> > > When we say “robustness” we mean “robust to changes in model weights”, which is a common goal of fingerprints. This property is useful when you want to detect when one model has been derived from another. On the other hand, in scenarios where the model weights are not expected to change, or when you wish to detect small changes in model weights, robustness may not be necessary nor desirable.

---

### Official Review · Reviewer_nUfp · 2025-10-31

**Soundness:** 3
**Presentation:** 3
**Contribution:** 2
**Rating:** 4
**Confidence:** 3

**Summary:**

This paper introduces ellipse signature for language model forensics. The core concept is that due to the final normalization and linear layers in modern language models, their output logits (e.g. log-probabilities) are constrained to lie on the surface of a high-dimensional ellipse specific to the model. The authors identify four distinctive properties: difficult to forge, intrinsic in current LM architectures, self-contained, and abundant. The paper presents an algorithm to extract a model's ellipse parameters from output samples and shows that, for large models, this extraction (and thus forgery of the signature) is computationally prohibitive. Furthermore, the authors propose an analogy to cryptographic message authentication.

**Strengths:**

- The idea of leveraging the intrinsic geometric constraint (the ellipsoid) imposed by standard model architecture as a signature is intuitive. The paper also shows a promising scenario for the practical application of the proposed signature.
- The authors identify four distinctive properties and provide theoretical/empirical evidences, especially for the forgery resistance that is the core concept of the paper.
- The authors keep comparing the ellipse signature with a wide range of existing methods throughout the paper, which gives readers a clear context to language model fingerprinting literature.

**Weaknesses:**

- Although the authors provide clear evidence for the distinctive properties of ellipse signature, they only analytically compare the ellipse signature to existing methods without any contrastive experiments, which makes readers hard to grasp the practical superiority of the ellipse signature.
- Both the concept of ellipse signature and the fitting algorithm are directly adopted from the prior work of Carlini et al, 2024. The contribution is somewhat limited since the paper mainly interprets the advanced properties of an existing method.
- While the ellipse signature is hard to forge shown in Figure 6, the paper does not show the trend with different scales of computation resources. Readers might concern that an adversary with sufficient resources might still extract the ellipse or generate on-ellipse outputs, given the polynomial-time complexity.
- Minor qualms: The references style is somewhat disordered. The authors are suggested to use a regular retrieval library and update the publishment for some accepted preprint literature.

**Questions:**

- The authors mention that the memory requirements of ellipse fitting makes GPU acceleration infeasible for ''sufficiently large models''. Can we use GPUs to accelerate smaller models (e.g. 1-10M) without surpassing the memory limitation?
- What is the actual resource requirements for the fitting algorithm run with CPUs and GPUs, respectively?
- Can the authors evaluate and discuss the trade-offs between the fitting algorithm and some alternative approximation methods mentioned in Section 3.3 ?

---

> ### Author Response · Authors · 2025-11-24
>
> Thank you for your constructive feedback\! We hope that we can address the concerns you raise in the following.
>
> ## Contrastive experiments with existing methods
>
> You bring up an important question about how to compare our method with existing methods. While writing the paper, we considered how best to show that the method works and explain each of the properties that sets the method apart from existing methods.
>
> First, to show that the method works, we conducted the experiment in Figure 3, which shows that the target model ellipse is detectably distinct from other model outputs. Since this experiment showed that it is easy to detect which model generated an output using the ellipse, we did not think an accuracy comparison to other methods would be particularly fair or helpful, since our method already would achieve 100% accuracy. Several other methods achieve similarly high accuracy, e.g., under the right settings the watermark from Kirchenbauer (2023) gets perfect accuracy as well. Thus we do not think that accuracy is a very relevant comparison point between our method and others.
>
> Next, we considered the properties that set our method apart, namely the “naturally occurring”, “self contained”, “compact and redundant” and “hard to forge” properties. Since these properties are binary (either a model has the property or it doesn’t) we couldn’t think of an experiment that would make sense here. For instance, it is not clear what experiment to run to measure whether a method “requires access to the model input”.
>
> We believe that adding a summary comparison table for the different properties would be helpful for readers. We have included this as Table 2 in the appendix. Would this be sufficient to address your concern here? Are there particular experiments that you would like to see?
>
> Lastly, we hope to convey that we are not claiming that ellipse signatures are superior to other methods (except for linear signatures, which only lack the hard-to-forge property), only that they are applicable under different scenarios and assumptions based on their properties. For this reason, we have added “we emphasize that signatures are not necessarily \\emph{better} than other methods, rather their properties make them more suitable in specific situations“ to the introduction.
>
> ## Known results
>
> You are correct that the fact that model outputs are on an ellipse and a method for fitting an ellipse to data is covered in Carlini et al. Indeed, the geometric properties of the layer norm and various ellipse fitting methods were known prior even to their work. Our contribution is the application of these ideas to language model signatures. For instance, we are the first to show that the uniqueness of the ellipse makes it useful for identifying the generating model (Figure 3). We investigate ellipsoid fitting methods for recovering model parameters, and are the first to quantify the effectiveness of these fitting methods for language models (Figure 4\) and how they improve with sample size (Figure 5). We also expand on Carlini’s ellipsoid fitting method by introducing the idea of using ellipsoid-specific methods for guaranteeing that the fitting method returns an ellipse (Section 3.1). We believe that, though the mathematical and algorithmic foundations are established, their application to model signatures is our novel contribution, and is worth dissemintatin to the scientific community.
>
> ## References
>
> Thank you for pointing this out. We have updated all published paper references.
>
> ## Questions
>
> **GPU speedups:** Since the GPU runs out of space so quickly, speed gains from GPU acceleration are modest, and only help for models that are already quite fast to fit to begin with (\<1min). We are running an experiment to demonstrate this and will share them soon.
>
> **Hardware:** We ran our experiments on 64 AMD EPYC 7643 48-core processors (footnote 4 on page 8\) We tried GPU acceleration on A100s.
>
> **Ellipse fitting approximations:** To approximate the ellipse fitting algorithm, we tried fitting the ellipsoid projected to lower dimensions, however since this operation sends points into the interior of the ellipse, the resulting fit does not accurately find the surface of the ellipse. We did not find any approximation method that could give us accurate estimations of the model parameters. For instance, the estimated singular values were off by up to an order of magnitude. This does not mean that there are no good approximation methods, but we have yet to find one.

---

> > ### Author Response · Authors · 2025-12-03
> >
> > Followup on the GPU experiment:
> >
> > - Pythia 14M finishes in 146 seconds on 8 cores (CPU), 27 seconds on GPU (A100 GPU) ($\approx5\times$ speedup)
> > - Pythia 70M takes >10 minutes on CPU (interrupted), and does not fit on an A100 GPU.
> >
> > As expected, small models do benefit from a (constant) speedup on GPU, but this benefit does not scale because larger model ellipse fitting does not fit on a GPU.

---

### Official Review · Reviewer_3e9D · 2025-11-03

**Soundness:** 1
**Presentation:** 3
**Contribution:** 3
**Rating:** 8
**Confidence:** 4

**Summary:**

The paper observes that logits output by a typical LLM lie on a high-dimensional ellipse. They argue that this ellipse serves as a sort of "signature" that uniquely identifies the model.

**Strengths:**

The idea of using the ellipse signature as an identifier for a model is very cool!
Even though I don't really see any applications (see the issues below with the ones they suggest), I think that the observation itself is interesting.

**Weaknesses:**

See "Questions" for more details.

1. It isn't a watermark at all, because one can't detect it from samples from the LLM. One would need to query the model to see if it is the given one, making it a totally different thing.

2. They make bold and misleading claims about the hardness of forging the signature. I shouldn't need to say that presenting a particular algorithm with high complexity does not mean that the problem is hard.

3. They never say what they actually mean by "forgery." I am highly suspicious of whether the ellipse signature is hard to forge at all. They "argue" that the signature is hard to *learn*, not hard to forge. These are very different. For instance, it could be possible to use a few samples from the ellipse to compute a new sample that is far from the others (which is presumably what they mean by "forgery"), even if it is hard to learn the parameters of the entire ellipse.

**Questions:**

1. I think that it's misleading to talk about this as a "watermark" without explaining the serious shortcoming of the method for that purpose: That this method requires access to the logits. If you want to use this for watermarking, you would need to be able to detect the signal in text. Maybe you can by just estimating logits from enough text samples, but that would presumably require an absurd amount of text that in particular contained many repeated tokens...
For example, you say In contrast [to watermarking papers], ellipse-based verification requires no changes on the provider side, and each generation step of the model independently bears the model’s signature, meaning that a generation of length 1 is sufficient to identify the source model." This is *completely false* because an actual token output by the model is not sufficient to detect the ellipse!

2. You repeatedly say that you "show" or "find" that the ellipse is unforgeable. This is ridiculous: When someone builds a new cryptosystem from a *standard, well-studied assumption*, even then they typically say that they "prove security under assumption X." You don't even state your actual computational assumption anywhere! It's just sort of implicit. This would be fine, except that you kind of masquerade as a cryptography result.

3. I found one place where you reference the difference between learning and forging, but I think it is incorrect. You say that "In fact, in the worst case Ω(𝑑^2) samples are required to find even a single new (not in the set of samples) point on the ellipse, since if the samples are in general position then for every point not in the samples there is an ellipse that includes the samples but not the point." But given just 2 samples in general position, I can learn the entire projected 1d ellipse containing those two points, and therefore forge a 3rd point.

---

> ### Author Response · Authors · 2025-11-24
>
> Thank you for your high level of engagement with our paper. We appreciate your constructive feedback and enthusiasm\! We want to be careful about not over-claiming or misleading.
>
> ## Not a watermark
>
> You raise the concern that the signature we propose is not a watermark. We agree that a signature differs from a watermark or fingerprint because it has different properties. We do not view this as a weakness, for the same reason that a bicycle is not worse than a car, just useful under different circumstances. Watermarks require additional implementation on the provider side, which makes them unsuitable in certain scenarios where a signature might prove useful; while signatures require logprob access, which would make a watermark more suitable for some APIs. To make this clearer, we have added “we emphasize that signatures are not necessarily *better* than other methods, rather their properties make them more suitable in specific situations“ to the introduction.
>
> ## Showing hardness
>
> You raise the concern that the fact that our ellipse fitting method is inefficient does not imply that there is no efficient signature forgery method. We agree that we do not mathematically prove the hardness of ellipse forgery—this was not our aim. Rather, we demonstrate that the best currently known techniques are not nearly efficient enough. We would be interested to see if future work comes up with fast ellipse forgery algorithms, but we do suspect that this might require major breakthroughs in computing. We have made this stance clearer in our paper revision.
>
> Revisions
>
> - Abstract \+ “using currently known methods”
> - Introduction \+ “We are not aware of any ellipse forgery method that avoids having to fit the ellipse, though we cannot at this time mathematically rule out the possibility. For this reason, we adopt the term \`\`forgery resistance'' rather than \`\`unforgeability''.”
> - Section 3 \+ “Ellipse signature forgery resistance therefore relies on the idea that, as far as we are aware, there is no known method to produce new points on an ellipse without first fitting an ellipse to the known points.”
>
> ## Forgery definition
>
> We define forgery on L234 and L424 (original draft), and our revision adds a more formal definition of forgery at the start of Section 3: Formally, given a set of model outputs\~$x\_1,\\ldots,x\_n$ and a black box signature verifying function\~$f$ such that $f(x\_i)=1$ for all $x\_i$, forgery requires producing a new output $\\hat{x}$ which passes verification, i.e., $f(\\hat{x})=1$.
>
> ## Forgery without fitting the ellipse
>
> We tried several techniques for signature forgery that do not require fitting the ellipse, but were unable to find one that would work. For instance, trying to fit a lower-dimensional ellipsoid section of the full ellipse is more computationally feasible, but requires sampling $O(n^2)$ co-planar points, which is vanishingly unlikely when the points are chosen at random from a high-dimensional space. Of course, just because our efforts failed does not mean someone else could succeed. Publishing this paper can bring this interesting problem to the attention of the community.
>
> ## Questions
>
> **Watermarks, text, and logits:** Indeed you are correct that detecting the ellipse in text is unlikely to work, and logprobs are essential for our method. We intentionally used the phrase “single generation step” rather than “single token” to indicate that we are detecting the signature in a single next-token distribution. We add a footnote (3) to clarify this in the introduction.
>
> **Claims of unforgeability:** We tried to be careful about statements regarding “infeasibility”, restricting our claims to \*practical\* infeasibility and saying that our system is only analogous to a cryptographic system. Indeed, our own findings show that the signature can be forged in polynomial time/samples/cost. We intentionally avoid the term “unforgeable”, opting instead to use “forgery resistant” as a more appropriate term.
>
> **1D ellipse forgery:** You are correct that in the 1D-case two points fully determine the ellipse, though in this case, there is no third point to forge since the ellipse only contains 2 points. In 2D, you need five coplanar points to determine the ellipse, which you are unlikely to sample from a language model given its high dimensionality.

---

### Author Response · Authors · 2025-12-03

Thank you to all the reviewers for your time and overall positive assessments!

Our paper introduces **ellipse signatures** as a novel forensic tool for language model identification, leveraging a fundamental geometric property: LM outputs lie on high-dimensional ellipsoids due to the final normalization layer.

**Reviewer 3e9D** praised our work as a "very cool idea," and our findings “interesting”. Their main concerns were about whether signatures can be considered watermarks (we clarified that they are in a different category) and worries that ellipses might not be hard to forge (we clarified we do not claim unforgeability, only forgery resistance, and answered their theoretical questions.)

**Reviewer nUfp** called our idea  "intuitive", our proposed application “promising”, and remarked that we provide clear context for understanding our method compared to others. Their main concerns were about novelty and empirical comparisons to other methods. We justified our choices for comparing our method to existing ones, and argued that, while the mathematical foundations of ellipse signatures are well established, the application to model signatures is novel.

**Reviewer Cfrc** called our method "novel and interesting," and our derivation  "theoretically sound." They required some clarification on whether or not the difficulty of inverting ellipses was a strength or weakness (strength). They also asked for a hypothetical application scenario, which we provided.

**Reviewer pMYU** called the ellipse signature a "powerful tool for model forensics," and complemented that our assumptions are “realistic”, and our derivation "direct". They had questions about ellipse signatures under different API assumptions (e.g., temperature), which we answered and added as an appendix in our paper revisions. They were concerned about the robustness of the signature, which we clarified is not a goal of our method.

There was some confusion among the reviewers about how to compare our method to watermarks and fingerprints. In response, we clarified that ellipse signatures are **not watermarks** but serve complementary purposes, and we justified our choices for how we compared our method to existing ones. In our revision, we added language to emphasize that signatures are not "better" than other methods, only that they're suitable for different scenarios, and added a comparison table to summarize.

In summary our contributions are

1. **A Novel Application**: First work applying ellipsoid geometry to model signatures.
2. **Practical Impact**: Provides verification protocol analogous to cryptographic MAC systems.
3. **Comprehensive Analysis**: Quantifies sample/computational complexity, of forgery, something prior work hasn't addressed.

The paper establishes a new type of signature for language models with desirable properties (forgery resistance) and practical applications for model forensics. Our work addresses a timely problem in LLM accountability. Our submission has been strengthened through reviewer feedback and we have made revisions addressing all reviewer concerns.

---

### Meta-Review · Area_Chair_MgcK · 2026-01-07

**Summary:**

The paper introduces the concept of an ellipse signature as a tool for language model attribution, grounded in the geometric observation that logits produced by modern LMs lie on the surface of a high-dimensional ellipsoid. The authors argue that this naturally occurring constraint can serve as a unique identifier for a model and frame the mechanism as a cryptographic-style message authentication code. Across the reviews, there is consensus that the central idea is novel, theoretically intriguing, and potentially impactful for the study of model fingerprinting and accountability.

**Reviewer Concerns:**

- Overstated claims: Reviewers note that the paper sometimes makes bold assertions about “unforgeability” without clearly stating assumptions or providing formal proofs. The distinction between learning and forging is blurred, and the cryptographic framing risks being misleading.
- Practical limitations: The method requires access to full logits, which are rarely exposed in real-world APIs. This undermines claims of applicability to watermarking or black-box attribution. The fragility of the signature under fine-tuning, quantization, or other model modifications is not addressed.
- Computational infeasibility: While forgery resistance is argued via high complexity, reviewers emphasize that polynomial-time adversaries with sufficient resources may still succeed. The scalability of the fitting algorithm remains a major obstacle.

**Reviewer Scores:**

Reviewer nUfp may have a chance to raise the score.

---

### Decision · Program_Chairs · 2026-01-26

Accept (Oral)